# Regulation of Kinase Signaling Pathways by α6β4-Integrins and Plectin in Prostate Cancer

**DOI:** 10.3390/cancers15010149

**Published:** 2022-12-27

**Authors:** Saara Koivusalo, Anette Schmidt, Aki Manninen, Tomasz Wenta

**Affiliations:** 1Disease Networks Research Unit, Faculty of Biochemistry and Molecular Medicine, Biocenter Oulu, University of Oulu, 90220 Oulu, Finland; 2Department of General and Medical Biochemistry, Faculty of Biology, University of Gdansk, 80-308 Gdansk, Poland

**Keywords:** α6β4-integrin, *ITGA6*, *ITGB4*, plectin, hemidesmosomes, prostate cancer, kinase

## Abstract

**Simple Summary:**

α6β4-integrins and plectin are the key structural components of hemidesmosomes that have been implicated in carcinogenesis and which are thus considered as potential cancer biomarkers and drug targets for anti-cancer therapies. In this review, we elaborate on the current knowledge of the kinase signaling pathways regulated by α6β4-integrins and plectin in the context of prostate cancer. We discuss an emerging scenario where hemidesmosomal α6β4-integrins and plectin function as tumor suppressors but adopt new oncogenic roles upon hemidesmosome disassembly in prostate cancer.

**Abstract:**

Hemidesmosomes (HDs) are adhesive structures that ensure stable anchorage of cells to the basement membrane. They are formed by α6β4-integrin heterodimers and linked to intermediate filaments via plectin. It has been reported that one of the most common events during the pathogenesis of prostate cancer (PCa) is the loss of HD organization. While the expression levels of β4-integrins are strongly reduced, the expression levels of α6-integrins and plectin are maintained or even elevated, and seem to promote tumorigenic properties of PCa cells, such as proliferation, invasion, metastasis, apoptosis- and drug-resistance. In this review, we discuss the potential mechanisms of how HD components might contribute to various cellular signaling pathways to promote prostate carcinogenesis. Moreover, we summarize the current knowledge on the involvement of α6β4-integrins and plectin in PCa initiation and progression.

## 1. Introduction

PCa is a complex disease that affects millions of men globally [1]. Although improvements in its early detection have reduced the mortality rate, PCa is still one of the leading causes of cancer-related death in men [2]. The 5-year survival rate for men with local PCa is around 98%, whereas the 5-year survival rate for men with metastatic PCa does not exceed 30% [3]. The most preferable metastatic target tissue for PCa are lymph nodes and bone. Since over 20% of PCa patients progress to metastatic disease, a better understanding of the molecular signatures differing between indolent and aggressive tumors is needed.

The metastatic cells that escape from their primary site and circulate in the bloodstream are exposed to many factors inducing cellular stress. These cells have to be able to endure shear forces, adapt to new surroundings at potential metastatic sites and avoid the surveillance and attack of the immune system [4]. Indeed, the invasion of cancer cells into nearby or distant tissue requires the reinforcement of many of their properties, including proliferation, migration, cell cycle progression and apoptosis-, anoikis- and drug-resistance. These features enable cancer cells to first disseminate and invade the primary site, eventually leading to intravasation and circulation in the bloodstream and finally extravasation and colonization of a new niche [5].

The extracellular matrix (ECM) has emerged as a critical driver of tumorigenesis [6,7]. It governs the reversible biochemical modifications that enable the achievement of specific epithelial-mesenchymal phenotype plasticity which is crucial for tumor progression and metastasis [8]. Thus, it is not surprising that the function of integrins, the major family of ECM receptors, is often perturbed in cancer cells [7,9]. Integrins form the basis for two main types of cell-ECM adhesions, the actin-linked focal adhesions (FAs) involved in cell migration, and the HDs that link to the intermediate filament network and form dynamic anchoring structures regulating cell adhesion and cellular differentiation [10]. Although FAs and HDs are distinct complexes, they seem to regulate each other via intricate signaling crosstalk that is not yet thoroughly understood [11,12,13]. The main core of HDs is formed by a heterodimer of α6- and β4-integrins which mediate the attachment of epithelial cells to the laminin-rich basement membrane (BM) and are often considered to function as tumor suppressors [14,15,16,17]. There are two types of HDs: highly organized type I HDs that are typical for squamous epithelia such as the skin epithelium, and less well-defined type II HDs that are found in simple epithelium [18]. The prostate epithelium is considered to express type II HDs and, therefore, in this review we mainly focus on the functions associated with type II HDs.

It has been reported that one of the most frequent events taking place during PCa pathogenesis is the loss of HDs [15,16,19,20]. In line with these observations, a strong reduction in β4-integrin levels is observed in HD-deficient PCa cell lines as well as in tumor tissues in PCa patients [19]. Moreover, the downregulation of *ITGB4* leading to HD disassembly is strongly correlated with PCa progression, metastasis, tumor stage, Gleason score, prostate-specific antigen (PSA) level and worse patient survival [19,21]. Furthermore, a reduced expression of β4-integrins was observed during the development of prostate intraepithelial neoplasia (PIN), and subsequent progression to prostate carcinoma [20]. Upon HD disassembly, α6-integrin and plectin are released from HDs and seem to play a key role in the activation and maintenance of tumorigenic properties in PCa cells [19,22,23,24]. To exercise these functions, the HD components activate or cooperate with many cellular kinases. Protein kinases are intracellular enzymes that catalyze the transfer of phosphate from ATP to other proteins and are considered central mediators of cellular signals with critical implications in the PCa progression as well. However, they are involved in the regulation of almost every cellular activity: proliferation, migration, apoptosis and cell cycle control [25]. The different kinase pathways are linked via complex signaling crosstalk and therefore the role of individual kinases and the associated signaling pathways are likely highly context-dependent. The role of kinase signaling cascades in PCa has been recently thoroughly reviewed [26,27,28]. While there are approximately 500 protein kinases in humans [29], in this review, we discuss those that have been reported to be associated with HD functions. 

## 2. Hemidesmosomal Components Regulate the Tumorigenic Properties of Cancer Cells

α6β4-integrins have been studied for decades and modulation of their function appears to be strongly linked to cancer progression. Both oncogenic and tumor-suppressing functions have been reported but the manner in which α6β4-integrin levels correlate with tumorigenesis and metastatic potential appears to be context-dependent and is thus still a point of debate. On the one hand, there are several papers showing the importance of upregulation of α6β4-integrins linked to cancer progression, especially in pancreatic, lung, breast and colon cancer [21,30,31,32]. On the other hand, downregulation of α6- or β4-integrins, associated with loss of HDs, has been positively correlated with poor PCa prognosis and increased metastatic potential of PCa cells [11,15,16,19,20,21,33]. Moreover, one of the most frequent events during PCa pathogenesis is the loss of HDs [15,16,20]. Recent data revealed that strong downregulation of β4-integrin causing HD disassembly is critical for the metastatic potential of PCa cells [19,33]. In line with these observations, reduced expression of β4-integrins was documented during the development of prostate intraepithelial neoplasia and its progression to prostate carcinoma [20]. These findings were validated by an extensive bioinformatics analysis of PCa patient cohorts [19,21]. Importantly, a recent analysis of a PCa tissue microarray (TMA) indicated that the loss of HDs correlated with a strong reduction in β4-subunit levels while significant levels of α6-integrins and plectin were retained in PCa patient tissues, as was seen for most of the PCa cell lines [19]. It was found that this scenario was particularly evident in phosphatase and tensin homolog (PTEN)-negative prostate tumors. In PTEN-positive prostate and breast cancer cells, loss of α6- or β4-integrin expression caused a strong reduction in the other heterodimer partner and plectin, while in PTEN-negative cells both the protein levels of the heterodimer partner and plectin were only modestly affected or even upregulated. Importantly, plectin expression was not only retained but also required for enhanced tumorigenesis [19]. In addition, mutations in locus rs12621278 within the *ITGA6* gene region and resulting in reduced *ITGA6* expression have been associated with increased susceptibility for the development of PCa, suggesting that, in some cases, HD-disruption via downregulation of α6-integrins might also promote PCa tumorigenesis in vivo [34]. 

Taken together, it seems that the function of α6β4-integrins in tumorigenesis can be different depending on whether the HDs are intact or not and this could be regulated by β4- or α6-expression levels, ECM composition, the genomic landscape of cancer cells as well as any signaling activities that regulate the activity and/or stability of α6β4-integrins.

### 2.1. Disassembly of HDs Is a Prerequisite for Initiation of Carcinogenesis Driven by Hemidesmosomal Proteins 

α6β4-integrins and plectin play a central role in the progression of several types of cancer. It has been postulated that HD disassembly occurs at and is a critical driver of the cancer initiation stage, especially in PCa. In normal tissues, laminin-332 has been shown to bind to α6β4-integrins, leading to HD assembly and inhibition of cell movement [35]. By contrast, in tumor tissues laminin-332 forms complexes with α3β1-integrins in actively migrating cells [36]. Growth factor receptor-mediated signaling is thought to regulate the integrin receptor switch from α6β4- to α3β1-integrins [37]. Although many studies have shown that laminin-332 expression is up- or downregulated in cancer, its expression levels in prostate, colorectal, and breast carcinoma are consistently strongly reduced in comparison to unchanged tissue [38,39,40]. Therefore, the reduction in laminin-332 levels or the stimulation of α3β1-integrin binding to laminin-332 may lead to loss of interactions between α6β4-integrins and laminin-332 resulting in HDs disassembly. Furthermore, it has been shown that laminin-332 levels are altered by epidermal growth factor (EGF) signaling. Mainiero et al. [41] found that activated EGFR induces phosphorylation of the cytoplasmic tail of β4-integrin, suggesting that the interaction of growth factor receptors with β4-integrin might be directly involved in HD disassembly. In support of such a hypothesis, α6- or β4-integrin knock-out in PTEN-negative PCa cells resulted in plectin upregulation and robust activation of epidermal growth factor receptor (EGFR) phosphorylation [19]. HD disruption might also be a consequence of the epigenetic modification of *ITGB4* expression. Recently, Wilkinson et al. [42] found that the *ITGB4* promoter region is epigenetically modified by hypermethylation in aggressive androgen receptor (AR)-independent prostate cancer cells. Such modifications caused the production of a shorter form of β4-integrin from the alternative transcription start site [42]. So, far, the role of this variant is unclear, but the shorter β4-integrin might be unable to pair with α6-integrin and thereby contribute to HD disassembly. 

Importantly, there is a growing body of evidence showing that an altered stromal microenvironment leads to modified interplay between stromal and epithelial cells which is a central factor promoting cancer progression [43,44]. In a normal prostate gland, the BM separates stromal from epithelial cells but even minor polarity defects, which are frequent during cancer initiation, may lead to compromised BM barrier function and aberrant interactions between the two compartments. Whether and how HD disintegration contributes to BM dysfunctions remains to be studied.

### 2.2. Regulation of Cell Proliferation 

When HDs are intact, the attachment of α6β4-integrins to laminin induces a conformational change enabling tyrosine phosphorylation of the cytoplasmic tail of β4-integrin [45,46,47,48]. This phosphotyrosine (pTyr) residue can be recognized and bound by proteins containing pTyr-binding domains, such as the Src homology-2 (SH2) or pTyr-binding (PTB) domain leading to the activation of the downstream signaling pathways. The adaptor protein Shc has both such domains, and it can interact with the pTyr-residue at the cytoplasmic domain of the β4-subunit [46]. The interaction between β4-integrin and Shc does not affect HD formation, but it is required for the phosphorylation of tyrosine residue in Shc [45]. The phosphorylated Shc can bind to growth factor receptor-bound protein-2 (Grb2), which in turn recruits a cytoplasmic molecule Son of Sevenless (SOS), leading to its translocation to the membrane where it promotes the activation of Ras small GTPase (Ras) [49,50]. Activated Ras recruits rapidly accelerated fibrosarcoma kinase (RAF) to the cell membrane, leading to RAF activation and dimerization. Activated RAF kinase phosphorylates and activates mitogen-activated protein kinase (MEK), which in turn phosphorylates and activates extracellular signal-regulated kinase (ERK). Finally, active ERK translocates to the nucleus, where it phosphorylates transcription factors that promote cell proliferation [51]. In principle, all elements of the Ras/RAF/MEK/ERK pathway can be overexpressed and/or activated in PCa, leading to further activation of the pathway [52,53,54].

It has been shown that α6β4-integrins are involved in the activation of the PI3K/AKT pathway. However, neither α6- nor β4-integrin can directly activate phosphoinositide 3-kinase (PI3K) because they lack the required binding domain in their cytoplasmic tail. However, they can initiate a signal leading to activation of the PI3K/AKT pathway, which is then passed on to other proteins involved in direct interaction with PI3K [55]. Insulin receptor substrates 1 and 2 (IRS-1/2) are cytoplasmic adapter proteins that do not contain intrinsic kinase activity but function by recruiting proteins to cell surface receptors, thereby assembling complexes that function as signaling intermediates linking the activation of PI3K and α6β4-integrins. Ligation of the α6β4-integrins promotes the tyrosine phosphorylation of IRS-1 and -2, increasing their association with PI3K [55]. Similarly, phosphorylated Shc recruited by the β4-integrin cytoplasmic tail not only activates the Ras/RAF/MEK/ERK pathway, but it can also trigger the PI3K/AKT pathway when accompanied by cytokine-inducible phosphorylated GRB2-associated-binding protein 2 (Gab2). This Shc-Grb2-Gab2 complex can activate PI3K [56]. Upon activation, PI3K phosphorylates phosphatidylinositol 4,5-bisphosphate (PIP2) to generate phosphatidylinositol (3,4,5)-trisphosphate (PIP3), which in turn recruits 3-phosphoinositide-dependent kinase 1 (PDK1) leading to AKT phosphorylation [57]. AKT, also known as protein kinase B, is a signaling nexus connecting several downstream pathways. These include the direct phosphorylation of a mammalian target of rapamycin (mTOR), which is highly expressed in PCa [58].

In PCa, several receptor tyrosine kinases (RTKs), including EGFR, ErbB2, MET and Ron, have been shown to phosphorylate β4-integrin, leading to the activation of the PI3K/AKT pathway and disassembly of HDs. Macrophage-stimulating protein (MSP) activates Ron, which in turn activates PI3K/AKT and, together with 14-3-3 proteins, phosphorylates α6β4-integrins, forming a complex and generating a feedback loop to further promote signaling [59,60,61,62,63,64]. Moreover, the activation of EGFR or protein kinase C (PKC) has been shown to lead to β4-integrin activation via phosphorylation of its S1356 and S1364 by ERK1/2 and ribosomal S6 kinase 1/2 (RSK1/2) [65]. This phosphorylation event prevents the ABD domain of plectin from interacting with β4-integrin, leading to HD disassembly. Plectin released from HDs binds to the receptor for activated kinase C1 (RACK1) and regulates PKC signaling critical for the MAPK/ERK signaling pathway, promoting proliferation [66]. This finding agrees with the observation that plectin is a Pro tumorigenic regulator of cancer cell proliferation in several cancers including PCa [19,24,67]. Plectin knock-down inhibited PCa cell growth both in vitro and in vivo [24]. Moreover, when released from HDs, plectin plays an important role in the modulation of proliferative signaling together with α6- or β4-integrins since plectin knock-out prevented the activation of the PI3K/AKT pathway in PTEN-negative PCa cells with disrupted HDs [19]. 

PCa cell proliferation can also be promoted through the activation of the AKT/IKK/NF-κB pathway. The nuclear factor kappa B (NF-κB) is normally localized in the cytoplasm as a complex with the inhibitory proteins IκB. IκB can be phosphorylated by the IκB kinase (IKK) complex, leading to its ubiquitination and degradation and release of NF-κB from the complex. NF-κB then translocates to the nucleus and activates the transcription of genes that promote cell survival [68,69]. Zahir et al. [70] reported that the cytoplasmic tail of β4-integrin is required for the activation of the Rac family of small GTPases induced by basal and epidermal growth factors. They found that constitutive activation of Rac through NFκB activation is sufficient to maintain the viability of breast tumor cells lacking functional β1- and β4-integrins [70]. Therefore, autocrine laminin 332–α6β4-integrins–Rac–NFκB signaling might be a pathway that is activated to sustain the proliferation of PCa cells when the contact with BM is lost and/or HDs disassembled. 

### 2.3. Regulation of Cell Migration

HD disassembly appears to be necessary for epithelial cell migration and tumor invasion (Figure 1) [13,71]. The known molecular drivers responsible for the disassembly of HDs are described in Section 2.1. It has been shown that upon disintegration of HDs, retraction fibers containing α6β4-integrins localize to the rear of actively migrating tumor cells and keratinocytes [72,73]. The exact mechanism by which HDs dynamics are controlled in cancer cells is not understood, but it has been suggested that specific phosphorylation and Ca^2+^ may be involved in this regulation. The specific amino acid residues phosphorylated within the β4-integrin molecule and their functional consequences for cell migration capacity were recently summarized in [74]. Activating the stimulus results in the phosphorylation of the cytosolic tail of β4-integrin, leading to the interruption of plectin binding to β4-integrin [75]. Dissociation of the plakin domain of plectin from the β4-integrin cytoplasmic tail depends on specific threonine phosphorylation at T1736 in the C-terminal region of β4-integrin by RSK1 or protein kinase D2 (PKD2) [76,77]. The phosphorylation of serine residues at S1356 and S1364 located in the connecting segment (CS) of β4-integrin is thought to be catalyzed by ERK and by RSK, respectively, and these events appear to contribute to HD disassembly [65]. After plectin detachment from HD, the exposed tyrosine and serine residues within the β4-integrin molecule become available for possible phosphorylation by RTKs, leading to the activation of PI3K, AKT and Ras homologous small GTPase (Rho)-mediated signaling pathways [30].

Although FAs and HDs are distinct complexes, they seem to dynamically regulate each other via intricate signaling crosstalk that is not yet thoroughly understood [11,12,13]. Several reports find that loss of HDs promotes cell migration by stimulating FA dynamics [11,78,79]. Microscopic analysis of exogenous and endogenous FA marker proteins showed higher FA dynamics and faster diffusion rates for FA-localized vinculin in and out of FAs in HD-depleted PCa cells. Moreover, several FA marker proteins formed larger FAs compared to cells with intact HDs [11]. Furthermore, upon HD disassembly plectin relocated to actin filaments with a concomitant increase in stress fiber formation, focal adhesion kinase (FAK) autophosphorylation and activation of the PI3K/AKT pathway. Another study reported upregulation and activation of FAs leading to increased actomyosin contractility and traction forces stimulating FAK phosphorylation and translocation of Yes-associated protein (YAP) into the nucleus [78,80]. Active YAP subsequently activates the Rho/ROCK/myosin light chain (MLC)-pathway at the transcriptional level, whereas FAK activates multiple signaling pathways promoting cell migration [78,81,82,83,84,85,86,87]. Furthermore, downstream of the PI3K/Rac/RhoA pathway, rhotekin plays an important role by interacting with S100 calcium-binding protein A4 (S100A4) to promote membrane ruffling, lamella formation and the generation of contractile forces facilitating tumor invasion [88,89,90]. Taken together, HD-mediated adhesion to laminin is essential for the suppression of FA maturation and cell spreading, possibly through a reduction in RhoA/ROCK/MLC signaling [78]. Therefore, RhoA remains inactive, contributing to low ROCK activity, thereby preventing myosin phosphorylation, and ultimately leading to a decrease in actomyosin contractility and traction forces [82].

### 2.4. Regulation of Cell-Death-Associated Pathways

Depending on the cellular context, integrins can promote either cell survival or cell death. As long as cells maintain a proper contact with the ECM, integrins initiate a signaling cascade that promotes cell growth and survival. Loss of integrin signaling in non-adherent cells leads to the inhibition of cell growth and the promotion of a specialized form of cell death known as anoikis (Figure 2) [30,91].

The most critical event and point-of-no-return in apoptotic signaling is the mitochondrial outer membrane permeabilization (MOMP) [92]. Under normal conditions, bcl-2-like protein 4 (Bax) forms a complex with the anti-apoptotic protein B-cell lymphoma-extra-large (Bcl-_XL_) in the mitochondrial outer membrane. Activation of apoptotic signaling induces phosphorylation of the BH3-only protein BAD that interacts with Bcl-_XL_ leading to the release of Bax from the complex. Bax release allows its oligomerization to form pores at the outer mitochondrial membrane [93]. MOMP leads to release of cytochrome c into the cytosol, triggering apoptosome formation and resulting in the activation of the caspase cascade [94].

In adherent cells, cell death via apoptosis can be triggered by intracellular events such as excessive DNA damage, ER stress or hypoxia. The pro-apoptotic signaling triggered by cellular stressors follows the same mitochondrial pathway as described for anoikis and is followed by the activation of caspases, including caspase-3 and -7. It has been shown that the cytoplasmic tail of β4-integrin is cleaved by activated caspase 3 in epithelial cells undergoing apoptosis [95].

Caspases can also be activated by death receptors [96]. Stegh et al. reported that after induction of extrinsic apoptosis by CD95 or TNF-R, initiator caspase-8 bound to and selectively cleaved plectin at Asp2395 [97]. Microscopic analysis revealed translocation of caspase-8 into plectin-rich foci resulting in complete cleavage of plectin upon stimulation of apoptosis in vivo. In another study, plectin was shown to be a target of caspase-3 and -7 cleavage in keratinocytes undergoing staurosporine-induced apoptosis [98]. Furthermore, plectin-deficient primary mouse fibroblasts showed impaired apoptosis-induced actin reorganization after CD95-mediated apoptosis suggesting that plectin is critical for this process [97,99]. Consistent with these findings, Ni et al. found that plectin plays a critical role in protecting podocytes from adriamycin-induced apoptosis and disruption of the F-actin cytoskeleton. Adriamycin treatment led to the downregulation of plectin levels followed by phosphorylation of Y1494 in β4-integrins and activation of FAK/p38 signaling pathway resulting in cell death. Activation of p38 subsequently decreased F-actin levels and inhibited actin polymerization resulting in a disrupted cytoskeleton [100]. In agreement with these findings, a recent study reported that a lack of β4-integrins in PTEN-negative PCa cells resulted in the upregulation of plectin and increased apoptosis- and anoikis-resistance [19]. Given that plectin is often upregulated in PCa cell lines and patients’ tissues representing an aggressive form of PCa it is possible that plectin upregulation might be a strategy adopted by PCa cells to avoid cell death enabling metastasis to other organs.

Metastatic PCa cells inhibit apoptotic pathways upon loss of ECM contact and avoid anoikis by several means. One of them is the redox-mediated pathway [101]. Reactive oxygen species (ROS) can directly oxidate proto-oncogene tyrosine-protein kinase Src at cysteine residues leading to its activation [102]. Activated Src can, in turn, phosphorylate EGFR which activates the AKT pathway [101]. AKT can phosphorylate BAD, leading to the inhibition of apoptosis [103,104]. EGFR can also phosphorylate BAD in a PI3K-independent manner by activating Ras/RAF/MEK/ ERK signaling cascade [105,106].

ROS signaling can disrupt cytoskeletal organization by oxidating actin, resulting in its disassembly and depolymerization and thereby promoting cell death [107]. In fibroblasts, overexpression of plectin was shown to stabilize the actin cytoskeleton, which in turn increased the number of actin filaments, thereby creating more binding sites for plectin [108,109]. A similar observation was made in PCa cells where plectin was released from β4-integrins upon HDs disassembly and associated with actin cytoskeleton. This approach is likely beneficial for cancer cells, as a stable actin cytoskeleton protects cells from ROS-induced damage and apoptosis. In line with these findings, structural studies by Spurny et al. showed that after oxidation, plectin becomes more stable due to the disulfide bond-mediated conformation change. Importantly, this protects not only actin but also vimentin from being oxidized. Plectin, by directly binding to vimentin, sterically blocks the access for ROS and thereby prevents collapse of the vimentin cytoskeleton [110]. Moreover, ROS have been shown to exert direct oxidation and activation of several kinases, including Src tyrosine kinase [111,112,113]. Oxidized Src at positions Cys245 and Cys485 can induce phosphorylation of the Tyr845 residue of EGFR, which leads to the activation of pathways mediated by Erk and Akt. It has been proposed that in PCa such a mechanism protects cancer cells from apoptosis [101]. 

Parathyroid hormone-related protein (PTHrP) is expressed by human prostate tissues and PCa cell lines [114]. PTHrP has been shown to increase α6β4-integrin expression at the cell surface via a non-canonical intracrine pathway which in turn facilitates growth factor receptor signaling by activating the PI3K/AKT pathway, leading to BAD phosphorylation and reduced apoptosis in an anchorage-independent manner. Moreover, intracrine PTHrP signaling results in the phosphorylation of glycogen synthase kinase-3 (GSK-3), leading to the inhibition of GSK-3 activity [114]. Since GSK-3 is known to phosphorylate MYC proto-oncogene (c-myc) targeting it into the ubiquitin-proteasome pathway, the inactivation of GSK-3 results in increased levels of c-myc and thus decreased apoptosis [115].

An additional escape route from apoptosis is through the activation of the PI3K/AKT/mTOR pathway. mTOR activates the mammalian target of the rapamycin complex 1 (mTORC1) that mediates protein synthesis by phosphorylating and activating ribosomal protein S6 kinase (p70S6), which in turn phosphorylates and inactivates BAD, thereby inhibiting apoptosis [116,117].

#### Role of p53 and PTEN in the Regulation of Hemidesmosomal-Components-Mediated Cell Death

p53 is one of the most prominent tumor suppressors, halting cell division upon the detection of DNA damage. Mutations in *TP53* gene encoding for the p53 protein are common in PCa [118,119,120,121]. The mechanisms by which the different mutated forms of p53 promote the development of PCa remain incompletely understood but it is evident that p53 mutants can modulate the activity of many signaling pathways including those associated with α6β4-integrins. Studies in colon and breast carcinoma have shown that p53 inhibits α6β4-integrins-mediated activation of AKT by promoting the caspase-3-dependent cleavage of this kinase [122,123]. Bachelder et al. postulated that the function of α6β4-integrins depends on the p53 status. In p53-deficient cells, α6β4-integrins promoted AKT-dependent survival of carcinoma cells, whereas in carcinomas expressing functional p53, α6β4-integrins stimulated the cleavage and inactivation of serine/threonine kinase AKT in caspase-3-dependent manner, indicating that p53 can inhibit survival signals emanating from α6β4-integrins [123]. 

Inactivation of the PTEN is among the most common genomic aberrations in PCa. Loss of PTEN function leads to the activation of the AKT pathway via enhanced phosphorylation of Thr-308 and Ser-473 residues in AKT [124]. PTEN is a dual-specificity lipid and protein phosphatase that has the ability to remove the phosphate group from both phospho-tyrosine and phospho-serine/threonine residues. It converts PIP3 into PIP2 and removes phosphate groups from the tyrosine residues of several substrate proteins. PIP3-to-PIP2 conversion results in the activation of PI3K and AKT, key factors regulating cell survival [125]. PTEN is involved in a complex network of interactions with p53 as it harms p53 stability while p53 enhances PTEN transcription. When PTEN is lost, the p53 pathway is strongly activated [126]. Recently, the loss of PTEN and HDs were shown to synergistically promote the tumorigenesis of PCa cells [19]. The dual loss of PTEN and HDs induced apoptosis- and anoikis-resistance via the activation of the EGFR/PI3K/AKT-cascade. A bioinformatic analysis of several PCa patient cohorts revealed a strong negative correlation between plectin and PTEN expression, suggesting that plectin was essential for tumorigenicity in PTEN-negative PCa cells with disrupted HDs [19].

### 2.5. Regulation of Cell Cycle

The involvement of HD components in cell cycle regulation has been documented (Figure 3). However, there are only a few papers specifically focusing on the involvement of α6β4-integrins and plectin in the regulation of cell cycle in PCa. Here, we compile these studies and correlate them with evidence for the role of HD components in the cell cycle in other cancer types.

The upregulation of plectin in PTEN-negative cells with disrupted HDs was associated with differential expression of signaling pathways regulating the cell cycle. Plectin knockout cancelled these effects, suggesting that in PTEN-negative PCa cells plectin when released from HDs regulates the cell cycle [19]. In line with this finding, Hong et al. showed that transmembrane protein 268 (TMEM268) induces an S-phase cell cycle arrest and disrupts cytoskeletal remodeling by interacting with β4-integrin [127]. They found that the depletion of TMEM268 led to polyubiquitination and subsequent degradation of β4-integrins, accompanied by a strong reduction in α6-integrin levels. This event was associated with a cell cycle arrest in the S-phase caused by the inactivation of the cyclin B1/cyclin-dependent kinase 1 (CDK1) complex [127]. TMEM268-KO gastric cancer cells had reduced viability, proliferation rate and adhesion, all contributing to observed reduced tumorigenicity in a xenograft mouse model. Moreover, the disruption of TMEM268-β4-integrin interaction in gastric cancer cells led to the displacement of plectin from the cytoplasmic tail of β4-integrins, leading to plectin binding to the actin cytoskeleton. 

The current data propose a model where, upon HDs disassembly, α6β4-integrins adopt new signaling functions and activate FAK/Src- and EGFR/PI3K/AKT/ERK-cascades, leading to the upregulation of cyclin D and the degradation of CDK inhibitors (p21 and p27) (Figure 3) [19,128,129]. Cyclin D is a sensor that integrates extracellular signals to regulate the cell cycle machinery as it plays a critical role in allowing the G1 to S phase transition [130]. The *CCND1* expression is upregulated by ERK, Ets-like factors, AP-1 and KLF8 [131,132]. Cyclin D binds to and activates cyclin-dependent kinase 4 and -6 (CDK4 and CDK6), which then phosphorylate retinoblastoma protein (Rb) leading to the dissociation of the Rb-E2F complex freeing E2F transcription factors to activate cyclin A expression and entry into the S phase. Furthermore, PI3K/AKT-mediated inhibition of GSK-3β increases the stability of cyclin D by preventing its degradation [133]. During the G2 phase, integrin-mediated signaling leads to the activation of FAK and polo-like kinase 1 (PLK1) [131]. PLK1 phosphorylates the activating phosphatase cell division cycle 25c (Cdc25c) that together with inhibiting kinase Wee1 regulate the activity of CDK1 and cyclin B at the late G2 phase. Cdc25c triggers a rapid activation of the CDK1 and cyclin B to allow entry to mitosis [131,134,135]. Moreover, plectin interacts with CDK1 to regulate the marked cytoskeletal rearrangements necessary for mitosis [136]. Upon phosphorylation of plectin at Thr4542 by the p34^cdc2^ kinase that is activated during the M phase, plectin rapidly dissociates from IFs and becomes diffusely distributed in the cytoplasm [136]. This leads to the disassembly of F-actin microtubule networks and causes the collapse of keratin IF networks in prophase [136]. 

The crosstalk between α6β4-integrins and the activation of p53 followed by the induction of p21 has been reported [137,138,139,140]. p21 inhibits cell cycle progression by binding to CDK2, cyclin D and cyclin E and by regulating the expression of Rb and E2F proteins [141,142]. Furthermore, members of the p53 family are central to the regulation of integrin expression. Bon et al. [143] revealed that homeodomain-interacting protein kinase 2 (HIPK2), which is involved in p53-mediated cellular apoptosis and regulation of the cell cycle, modulates *ITGB4* expression [143,144]. HIPK2 depletion activates tumor protein p63 and p73 (TAp63 and TAp73) functions, leading to p53-dependent activation of *ITGB4* transcription [145]. TAp73, a tumor suppressor inducing apoptosis and cell cycle arrest [146], has been shown to bind the *ITGB4* promoter region to stimulate *ITGB4* transcription [145]. In addition to transcriptional regulation, β4-integrin levels can be altered by netrin-4 [147] and human leukocyte antigens (HLA), both of which have been implicated in cell cycle progression [148]. However, the mechanistic details of these interactions are still elusive.

It has been demonstrated that the knock-down of *ITGA6* upregulates cyclin-dependent kinase inhibitor p27 in the G1/S phase and transcription factor E2F1 that subsequently downregulates cdc25a and cyclin E/CDK2, leading to a deceleration of cell cycle progression by inhibition entry into the S phase [149]. Activated cyclin E enables passage through the G1/S phase by binding to CDK2, leading to phosphorylation and inactivation of Rb and release of E2F transcription factors [150]. Furthermore, the downregulation of α6-integrins was associated with upregulation of checkpoint kinase 1 (chk1), resulting in cdc25a downregulation and inhibition of Rb phosphorylation [149]. chk1 phosphorylates cdc25a, generating a binding site for inhibitory 14-3-3 protein. The binding of 14-3-3 regulates the S-phase checkpoint in response to genotoxic stress induced by DNA-damaging agents or irradiation [150,151,152]. These data suggest that α6-integrin depletion prevents cells from passing through the G1/S. In line with this finding, Hu et al. showed that depletion of α6-integrin affected arrested cells in the G2/M checkpoint only when the cells had been treated with infrared radiation. α6-integrin expression was associated with elevated DNA repair activity in irradiated cells, and this correlated with the activation of PI3K/AKT- and MEK/ERK-signaling pathways [153].

Plectin knock-out was associated with the downregulation of *CCND1* [154]. By interacting with breast cancer susceptibility protein (BRCA2), which is involved in S phase checkpoint activation and centrosome duplication, plectin regulated the perinuclear positioning of the centrosome during the S phase [155]. Furthermore, at the beginning of the M phase, plectin is phosphorylated by CDK1, resulting in cytoskeletal rearrangements and the dissociation of plectin from the IF network to promote centrosome positioning [155,156]. In line with this finding, plectin downregulation was associated with centrosome dislocation and abnormal nuclear morphology [155]. Interestingly, the depletion of a specific plectin isoform, 1c, led to mitotic spindle defects in primary basal keratinocytes [157]. Plectin appears to be involved in the maintenance of genomic stability and proper cell divisions. Moreover, studies using plectin-binding peptides resembling a natural degradation product of cyclin D2 revealed that cells treated with the peptide became stuck at the G1/S checkpoint, resulting in collapse of the cytoskeleton and cell death [158]. Accordingly, Perez et al. recently showed that a monoclonal antibody (1H11) specifically targeting a cancer-specific plectin isoform inhibited the plectin activity and caused cell cycle arrest at the G0/G1 checkpoint in the antibody-treated cells [159].

Intriguingly, HDs can directly affect the organization of the nucleus. Using a yeast two-hybrid screen assay, nesprin-3 was identified as plectin interacting protein binding to its plakin domain [160]. Nesprin-3 is an integral transmembrane protein of the outer nuclear membrane, and it has been implicated in flow-induced centrosome polarization. Plectin function can also be modified by phosphorylation by PKC and cyclin adenosine monophosphate (cAMP)-dependent protein kinase A (PKA), which regulates the interaction between plectin and an IF protein vimentin as well as the association of plectin with laminin B [161]. Recently, Serres et al. showed that vimentin is recruited to the cell cortex in mitotic cells in a plectin-dependent manner [162]. They found that during the interphase, plectin orchestrated the assembly of cytoplasmic networks by interlinking IF, microtubules and actin filaments into a cortical layer after entry into mitosis [136,162]. The depletion of plectin caused loss of vimentin at the cortex and led to vimentin accumulation in cytoplasmic bundles and an increase in the thickness of the cortical networks. Furthermore, plectin depletion also affected the mechanical properties of the actin cortex, resulting in increased cortical tension driving mitotic rounding. Hence, plectin seems to be critical for vimentin recruitment to the cell cortex in mitotic cells, thereby regulating thinning of the cortical actin network and cortical tension [162].

## 3. Drugs Targeting Hemidesmosomal Components as Potential Anti-Cancer Therapy 

Because the key HD proteins, α6β4-integrins and plectin, play multiple roles in both normal cell differentiation and tumor progression, their use as drug targets in cancer therapy is a very challenging task. Their involvement in the development of multidrug resistance and their role in cell adhesion and signaling makes this aim even more complicated [31,163]. The upregulation of *ITGA6* is central to the development of cisplatin-resistant ovarian cell lines, and drug-resistant tumors in patients [163]. Furthermore, increased expression of α6-integrin correlates with drug resistance to leukemia [164]. Cancer-specific plectin isoform was reported to mitigate drug sensitivity and reduced plectin levels correlated with better efficacy for several drugs such as sorafenib [34,159]. 

The activity of α6β4-integrins might be dynamically modulated by miRNAs that inhibit translation or direct mRNA degradation by binding to target mRNAs, often at their 3′-untranslated regions (3′-UTRs) [121]. miRNAs can regulate multiple target genes and play multidimensional roles in cancer as tumor suppressors or oncogenes, which has led to increased interest in using miRNAs as therapeutic tools [165]. *ITGA6* is a direct target gene for miR-25 that has been shown to regulate the invasiveness of PCa cells. Zoni et al. found that miR-25 is expressed at low levels in normal and transformed prostate stem cells but is rapidly upregulated upon luminal differentiation, leading to the downregulation of α6- and αV-integrins and thereby inhibiting invasive properties and PCa cell extravasation of tumor stem/progenitor cell subpopulations [166].

*ITGA6* expression is suppressed also by another miR, miR-143-3p. The downregulation of *ITGA6* in gallbladder carcinoma by miR-143-3p decreased tumor growth by reducing *PLGF* expression in a PI3K/AKT-dependent manner [167]. In addition, p53-induced miR-30e-5p is involved in the modulation of *ITGA6* levels. Laudato et al. found that miR-30e-5p inhibits colorectal cancer invasion and metastasis, and upregulates p21 and p27, leading to G1/S cell cycle arrest by targeting *ITGA6* and *ITGB1* [168]. Furthermore, aberrant expression of *ITGA6* might be induced by the miR-92a [169] and miR-29-3p, which negatively regulate the transcription activator of *ITGA6*/*ITGB1* by SP1 [170]. 

The most studied approach for interfering with integrin functions is to use antibodies. The activity of α6-integrin can be blocked by monoclonal GoH3 and J1B5 antibodies, which are specific for α6-integrin and selectively inhibit both adhesion and signaling functions, whereas another monoclonal α6-targeting antibody J8H efficiently inhibits the processing of α6-integrin to the truncated form ItgA6p without affecting adhesion to laminin-511 in vitro [171,172,173]. The J8H antibody specifically recognizes and binds to the extracellular domain of full-length α6-integrin and might be useful as a less cytotoxic strategy to improve current therapies for patients with advanced PCa and extensive bone metastasis. In support of such a hypothesis, Landowski et al. reported that J8H antibody-mediated blocking of α6-integrin cleavage inhibited PCa progression and metastasis to the bone in mice. Such additional treatment enhancements could thus help with disease control without inducing significant systemic toxicity [171].

Other strategies have been developed to block plectin activity. Chen et al. designed plectin-1 targeting dual-modality nanoparticles (plectin-SPION-Cy7). Such particles have a high affinity for cancer-specific plectin (CSP) and accumulate within tumors but not in normal pancreatic tissue [174]. Similarly, Bausch et al. generated a CSP-targeting agent to visualize tumor localization and distribution using single-photon emission/CT (SPECT) [175]. Iron oxide nanoparticles can be applied for magnetic techniques such as magnetic resonance imaging (MRI), magnetic particle imaging (MPI), and magnetic hyperthermia. Bovine serum albumin superparamagnetic iron oxide nanoparticles (BSA·SPIONs) conjugated with near infra-red fluorescent dyes, Cy5 or Cy7, and monoclonal anti-plectin antibody specifically were shown to associate with CSF-plectin in pancreatic cancer cells and mice tissue [174]. Moreover, the metallodrugs have been tested. Plecstatin-1 is an organoruthenium drug that specifically targets plectin [176,177,178]. In colon and breast cancer cell lines, plecstatin-1 induced G0/G1 checkpoint arrest, leading to increased phosphorylation of eIF2α, increased ROS levels and reduced spheroid growth in vitro and tumor growth in mice in vivo [176,177,178]. 1H11 monoclonal antibody that selectively binds to cancer-specific plectin with a high specificity has been generated [159]. Treatment with this antibody resulted in JAK2-STAT3 pathway inhibition, upregulation of p21 and p27 and differential expression of EMT markers, leading to cell cycle arrest, decreased migration, proliferation, and tumor growth of ovarian tumor cells. The 1H11 antibody has remarkably low toxicity and it has shown synergistic activity with currently used chemotherapeutics. The administration of 1H11 together with cisplatin resulted in significantly greater tumor inhibition than cisplatin alone [159]. Taken together, plectin appears to be a valuable therapeutic target to improve current therapies.

Summing up, current approaches focusing on simultaneous targeting of integrin-mediated adhesion and signaling in most cases result in high toxicity. This is especially evident when targeting α6β4-integrins, which have tissue-limited distribution in the epithelium but which is critical to maintain the integrity of the epithelium [179,180]. Thus, targeting the downstream effectors of integrin signaling could be considered to reduce potential toxic effects, e.g., by using PI3K inhibitors that decrease integrin α6-integrin expression, thereby reducing tumorigenic potential without completely blocking integrin functions [181].

## 4. Conclusions and Future Directions

The role of HDs in carcinogenesis is complex and still incompletely understood. However, the oncogenic functions of plectin and α6-integrin upon HD disassembly appear to be consistent in many cancer types. β4-integrin, due to its unique cytoplasmic tail and ability to induce multiple signaling pathways, is more controversial and its cancer-related functions are likely context-dependent and vary between different cancer types. Therefore, the role of α6β4-integrins might vary significantly even within different PCa subtypes, depending on the genomic background, and the functions reported in other cancer types such as breast, colon, and pancreatic cancer need to be confirmed using a PCa model. However, the reduction of β4-integrin levels leading to HD disassembly seems to be a key tumor-promoting factor in PCa. More knowledge about the exact mechanisms, how and when HDs disassemble needs to be accumulated. Moreover, careful mechanistic studies are needed to understand how the lack of β4-integrin drives PCa tumorigenesis and how common this is in PCa and other tumor types in general. 

Multiple studies have emerged implicating plectin as an oncogenic factor due to its involvement in the regulation of the tumorigenic properties of cancer cells. In several cancer types, including pancreatic and ovarian cancers, plectin has been shown to relocate from the cytoskeleton to the cell surface. The increased level of plectin has been positively correlated with PCa progression and metastasis [19,24]. Moreover, the CSP cell surface translocation has been proposed as a prognostic biomarker in several cancer types and serves as one of the most promising targets for improved anti-cancer therapies [23]. Unfortunately, the mechanisms regulating plectin translocation are still elusive. It is also unclear which plectin isoform undergoes this cancer-specific translocation and how this contributes to tumorigenesis. 

Finally, one key feature of PCa pathogenesis is the dependence on androgen signaling. How α6β4-integrins and plectin may regulate or contribute to AR-signaling remains to be investigated in detail in future studies. 

## Figures and Tables

**Figure 1 cancers-15-00149-f001:**
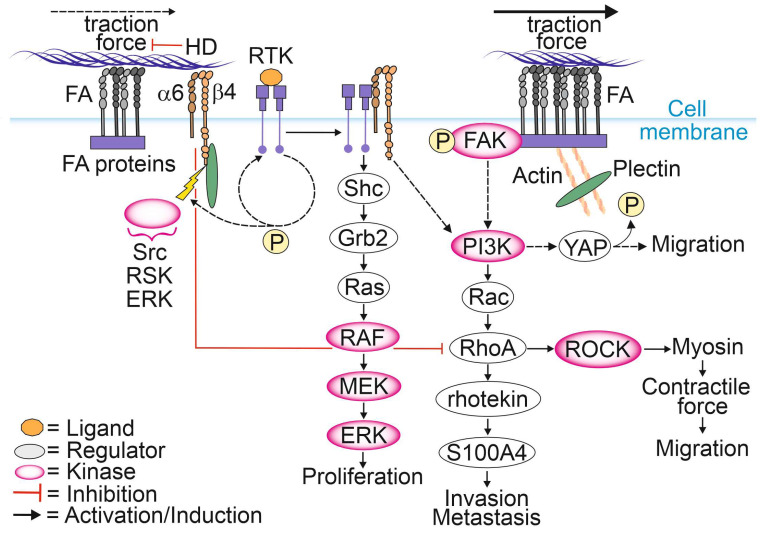
Involvement of α6β4-integrins and plectin in the regulation of cell proliferation and migration.

**Figure 2 cancers-15-00149-f002:**
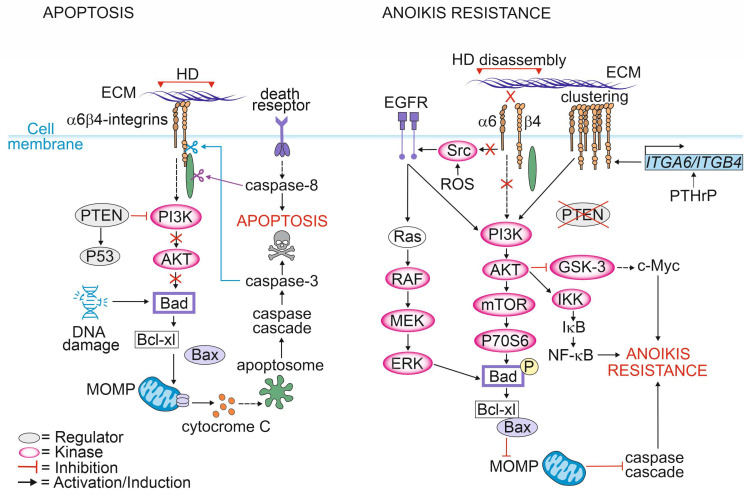
Mechanisms of apoptosis resistance induced by hemidesmosomal proteins.

**Figure 3 cancers-15-00149-f003:**
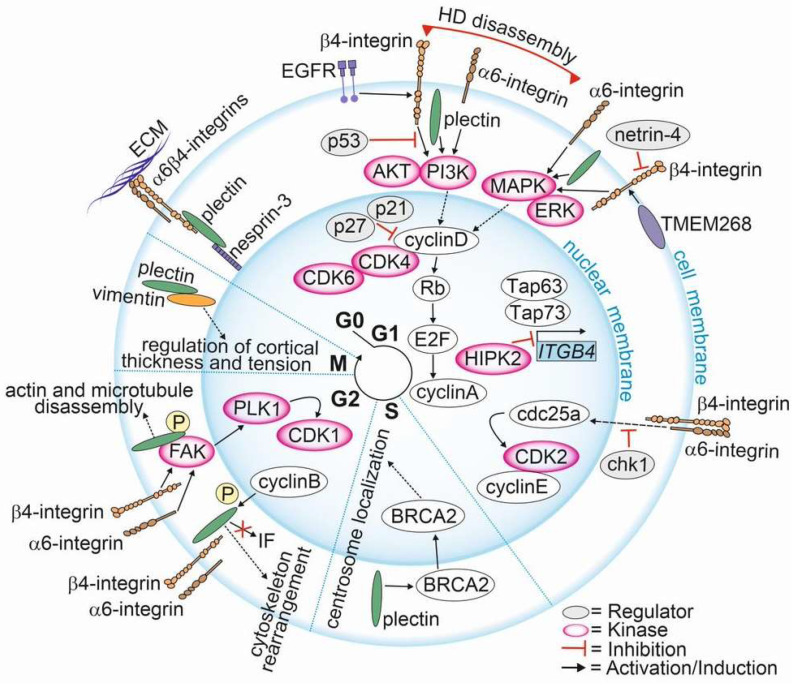
Involvement of hemidesmosomal proteins in cell cycle regulation.

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
