# Peer review of "Regulation of Kinase Signaling Pathways by α6β4-Integrins and Plectin in Prostate Cancer"

_cancers, 2022, doi:10.3390/cancers15010149_

Round 1

Reviewer 1 Report

Overview and general recommendation: 

In the manuscript, the authors introduce that the adhesive structures- hemidesmosomes (HDs) are involved in cell adhesion and cellular differentiation. HDs are formed by α6β4-integrin, a transmembrane heterodimer. During prostate cancer’s (PCa) progress, the expression level of β4-integrin is reduced and the expression level of β4-integrin remains similar or even improved, thus regulating different signaling pathways, which promote the tumorigenesis of PCa. The authors introduce and discuss the role of α6β4-integrin-mediated signaling pathways in PCa. 

First they show that HDs are formed by α6β4-integrins and in PCa progression, loss of HDs is one of the most important events. α6β4-integrins can regulate many cellar kinases to maintain the tumorigenic properties of PCa cells. Then the authors discuss how tumorigenesis, cell proliferation, cell migration, cell death and cell cycle are regulated by α6β4-integrins during PCa progress. They also discussed the potential of α6β4-integrins-targeted drugs in therapy of PCa. 

I find the paper is well organized. The author performed detailed background research and all the parts are organized in a logic way. The figures are quite clear to show the model proposed by the authors. But it is essential to include the references in the manuscript.  

Major comments: 

  1. Since the authors talk about the role of kinase signaling pathways by α6β4-integrins and plectin in cancer cells, I suggest the authors the authors add a section which describes the regulation of tumorigenesis. 
  2. In 2.3 Regulation of cell migration, the authors describe that Fas and HDs work coordinately in regulating cell migration. I suggest the authors add some detailed information about how cytoskeleton dynamics is regulated by kinase signaling pathway by α6β4-integrins. 
  3. In 2.4. Regulation of cell death associated pathways, the ROS signaling part seems not very closely connected to the α6β4-integrins and plectin. I suggest the authors add more information to strengthen the relationship of ROS signaling and α6β4-integrins and plectin. 

Author Response

Reviewer #1:

We are grateful for your insightful comments and constructive criticism of our review paper.

I find the paper is well organized. The author performed detailed background research and all the parts are organized in a logic way. The figures are quite clear to show the model proposed by the authors.

We are glad that the reviewer found our review interesting and believes that this work may be useful to a wide audience.

  1. Since the authors talk about the role of kinase signaling pathways by α6β4-integrins and plectin in cancer cells, I suggest the authors the authors add a section which describes the regulation of tumorigenesis. 

We thank for the comment. The purpose of our work was to summarize and compile information on the kinase-mediated regulation of prostate cancer formation and progression by hemidesmosomal components. Thus, in our review, we focused on presenting new findings and links providing comprehensive insights into this process. The papers regarding the general function of kinases in prostate cancer progression have been recently published (PMID: 35335890, PMID: 35326402 and PMID: 22761715). We have now added information and references concerning the role of kinases in more general in lines 39-51 and 76-85.

  1. In 2.3 Regulation of cell migration, the authors describe that Fas and HDs work coordinately in regulating cell migration. I suggest the authors add some detailed information about how cytoskeleton dynamics is regulated by kinase signaling pathway by α6β4-integrins.

In this review, we tried to fill a gap in the current state of knowledge. The review paper regarding the involvement of α6β4-integrins in regulation of migration and cytoskeleton dynamic has been recently published (PMID: 34523678). Thus, to avoid repetition of similar information we included in our paper appropriate references - lines 230-236.  

  1. In 2.4. Regulation of cell death associated pathways, the ROS signaling part seems not very closely connected to the α6β4-integrins and plectin. I suggest the authors add more information to strengthen the relationship of ROS signaling and α6β4-integrins and plectin.

We thank for the comment. In current version we included additional explanations to strengthen the relationship of ROS signaling and hemidesmosomal components (lines 328-337).

Reviewer 2 Report

Koivusalo et al. present a review article on kinase signaling pathways in hemi-desmosomes, mainly involving α6β4-integrins and plectin. This was discussed in the context of prostate cancer. The authors should be commended for a very comprehensive review which covers both basic science and translational implications for prostate cancer patients.

Three minor comments:

-Both Fig 1 and Fig 2 are not referenced in the body of the text

-Figure 2 is difficult to understand. Hard to know what the difference in signaling is showing.

-Should G2 phase in Figure 3 really be CDK1 interacting with CyclinB? 

Author Response

Reviewer #2:

We are grateful for your insightful comments and constructive criticism of our review paper.

Koivusalo et al. present a review article on kinase signaling pathways in hemi-desmosomes, mainly involving α6β4-integrins and plectin. This was discussed in the context of prostate cancer. The authors should be commended for a very comprehensive review which covers both basic science and translational implications for prostate cancer patients.

We are glad that the reviewer found our review interesting and believes that this work may be useful to a wide audience.

1. Both Fig 1 and Fig 2 are not referenced in the body of the text.

We thank for the comment. The figures are now referred in the main text.

2. Figure 2 is difficult to understand. Hard to know what the difference in signaling is showing.

To clarify the difference between the left panel showing the apoptosis pathway and the right panel presenting anokis resistance, we modified the figure accordingly. We hope the current version of this figure is more straightforward.

3. Should G2 phase in Figure 3 really be CDK1 interacting with CyclinB? 

According to the current work (PMID: 32120135 and PMID: 28591578), CDK1 and cyclin B are specifically activated to allow entry into mitosis at the G2 phase. We clarified this information in our paper (line 416-417).

Round 2

Reviewer 1 Report

In the manuscript, the authors introduce kinase signaling pathways related to the adhesive structures- hemidesmosomes (HDs) in cell adhesion and cellular differentiation. They add some complementary information. I find that the authors have put considerable effort into addressing the reports of the referees. As a result the paper is very much improved and I have no problem in recommending it for publication.